# Online Detection of Hydrogen Fluoride under Corona Discharge in Gas-Insulated Switchgear Based on Photoacoustic Spectroscopy

**DOI:** 10.3390/s24092806

**Published:** 2024-04-27

**Authors:** Liujie Wan, Xiaohe Zhao, Kang Li

**Affiliations:** 1School of Cable Engineering, Henan Institute of Technology, Xinxiang 453003, China; 2Henan Key Laboratory of Cable Structure and Materials, Xinxiang 453003, China; zxh2009008@hait.edu.cn; 3School of Electrical Engineering and Automation, Henan Institute of Technology, Xinxiang 453003, China; 4Institute of Electrical Engineering, Chinese Academy of Sciences, Beijing 100190, China; likang07@mail.iee.ac.cn

**Keywords:** photoacoustic spectroscopy, GIS, partial discharge, hydrogen fluoride

## Abstract

Internal discharge and overheating faults in sulfur hexafluoride (SF_6_) gas-insulated electrical equipment will generate a series of characteristic gas products. Hydrogen fluoride (HF) is one of the main decomposition gases under discharge failure. Because of its extremely corrosive nature, it can react with other materials in gas-insulated switchgear (GIS), resulting in a short existence time, so it needs to be detected online. Resonant gas photoacoustic spectroscopy has the advantage of high sensitivity, fast response, and no sample gas consumption, and can be used for the online detection of flowing gas. In this paper, a simulated GIS corona discharge experimental platform was built, and the HF generated in the discharge was detected online by gas photoacoustic spectroscopy. The absorption peak of HF molecule near 1312.59 nm was selected as the absorption spectral line, and a resonant photoacoustic cell was designed. To improve the detection sensitivity of HF, wavelength modulation and second-harmonic detection technology were used. The online monitoring of HF in the simulated GIS corona discharge fault was successfully realized. The experimental results show that the sensitivity of the designed photoacoustic spectroscopy detection system for HF is 0.445 μV/(μL/L), and the limit of detection (*LOD*) is 0.611 μL/L.

## 1. Introduction

SF_6_ gas possesses many excellent characteristics, such as high dielectric strength, strong chemical inertness, and low toxicity, which significantly improve the reliability and stability of equipment and reduce the labor intensity of the routine inspection and operation maintenance of equipment, and thus, SF_6_ has been widely used in GIS [1,2].

Few decomposition products are generated during the normal operation of GIS. In the case of a discharge or overheating fault in GIS, SF_6_ will decompose and produce a series of low-fluorine sulfides (S_x_F_6−x_, 1 ≤ x ≤ 5). These low-fluorine sulfides will repolymerize into SF_6_ after the faults disappear. However, when trace amounts of water or oxygen are present in SF_6_, these low-fluorine sulfides will react with the impurities to produce more compounds, such as HF, SO_2_, H_2_S, CO, CF_4_, CS_2_, SO_2_F_2_, SOF_2_, etc. [3].

Van Brunt [4] from the National Bureau of Standards of the United States proposed a three-region model to explain the SF_6_ decomposition mechanism during discharge, which is widely recognized. In his theory, the SF_6_ discharge region can be divided into three zones, as shown in Figure 1: the glow zone, ion migration zone, and main gas chamber zone. In the presence of water and oxygen, SF_6_, water, and oxygen are dissociated under electron collision in the glow region to form low-fluorine sulfide, O, OH, and F particles. These particles will react in the glow region, and the reaction processes are shown in Formulas (1)–(8) [3].
(1)$$SF_{5}+O={SOF}_{4}+F$$
(2)$$SF_{5}+OH={SOF}_{4}+HF$$
(3)$$SF_{4}+O={SOF}_{4}$$
(4)$$SF_{4}+OH={SOF}_{4}+HF$$
(5)$$SF_{3}+O={SOF}_{2}+F$$
(6)$$SF_{3}+OH={SOF}_{2}+HF$$
(7)$$SF_{2}+OH={SOF}_{2}+HF$$
(8)$$SF+O+F={SOF}_{2}$$

In the ion migration zone, some ions react with the compounds produced in the glow zone, and the reaction in this zone has little influence on the final product. The main reaction process is shown in Formulas (9)–(11).
(9)$$SF_{6}+{SOF}_{4}=\cdot {SOF}_{5}+{SF}_{5}$$
(10)$$SF_{6}+{SO}_{2}=\cdot SO_{2}F+2HF$$
(11)$$S+\cdot O_{2}=S{+O}_{2}$$

When the stable products produced in the glow zone are diffused to the main gas chamber zone, they will react with the water and oxygen in the main gas chamber region to generate more stable products. The main reaction process is shown in Formulas (12)–(15).
(12)$$SF_{2}+O_{2}=SO_{2}F_{2}$$
(13)$$SF_{4}+H_{2}O=SOF_{2}+2HF$$
(14)$$SOF_{2}\cdot +H_{2}O=SO_{2}+2HF$$
(15)$$SOF_{4}+\cdot H_{2}O=SO_{2}F_{2}+2HF$$

According to the discharge mechanism of SF_6_, HF is one of the main decomposition products. HF is very active. It can strongly corrode metal materials, which brings great security risks to the safe operation of GIS. IEC60480-2020 clearly states that SF_6_ will produce HF under discharge and high-temperature faults, and it stipulates that the maximum allowable concentration of HF in SF_6_ gas insulation is 25 μL/L [5].

The discharge and overheat faults in GIS will produce HF, so HF is a very important fault characteristic gas in GIS [6]. The chemical properties of HF are very active, which leads to HF’s existence for a short time. Therefore, it is difficult to detect HF in the offline detection of gas in GIS.

The traditional HF gas detection methods mainly include infrared absorption spectroscopy, detection tubes, and carbon nanotube gas sensors. Infrared spectroscopy has wide spectral lines, dense absorption peaks, and poor anti-interference ability, and it is easily affected by environmental gases and makes it difficult to detect trace gases. The detection tube method can realize the detection of HF gas by using the principle of chemical color reaction, but it is easily affected by the environment and there is gas interference. The accuracy of the carbon nanotube method is low, and it makes it difficult to realize online detection. Fourier transform infrared absorption spectroscopy can also be used to detect HF gas, but its volume is large, the sensitivity is low, and it is not suitable for online monitoring [7,8]. Tunable Diode Laser Absorption Spectroscopy (TDLAS) has also been used to detect HF [8,9,10]. In recent years, the detection of HF gas by photoacoustic spectroscopy has been rare. Teemu Tomberg et al. adopted the cantilever beam-enhanced photoacoustic spectroscopy technique based on an optical parametric oscillator to detect the strong absorption of spectral lines of HF at 2475.8836 nm and found that the *LOD* of HF was below nL/L [11]. All of these studies focused on HF standard gases. At present, there is no research on the online detection of HF gas generated during simulated GIS discharge.

Gas photoacoustic spectroscopy is a non-background gas detection technology with high accuracy, good long-term stability, and high sensitivity. Resonance photoacoustic spectroscopy can detect flowing gas [12,13,14,15,16]. Therefore, resonance photoacoustic spectroscopy was used in this research to monitor the HF generated during a simulated discharge fault. The research results are of great significance for the safe operation of GIS.

## 2. Principle of Resonant Photoacoustic Spectroscopy Detection Technology

Gas photoacoustic spectroscopy is a kind of detection technology based on the photoacoustic effect, which is caused by the periodic non-radiative relaxation (thermal effect) caused by the absorption of changing light energy by gas molecules. The sound pressure generated by gas molecules in photoacoustic cells can be expressed as the wave formula [17,18,19,20].
(16)$$\nabla^{2}p-c^{-2}\frac{\partial^{2}p}{\partial t^{2}}=-\frac{\gamma -1}{c^{2}}\frac{\partial H}{\partial t}$$

In Formula (16), *p* represents the sound pressure of the gas, and *c* represents the sound velocity in the gas; *γ* = *C_P_*/*C_V_*, which represents the specific heat ratio of the gas; *H* represents the thermal power density generated by the gas absorption-modulated light energy. If the incident light intensity is *I*, *H* = *αI*, *α* is the absorption coefficient of the gas molecule.

In a cylindrical coordinate system, considering gas heat conduction loss and viscosity loss, the amplitude *A_j_*(*ω*) in normal mode *j* can be expressed as follows:(17)$$A_{j}\left(\omega \right)=-\frac{i\omega }{\omega_{j}^{2}}\;\frac{\frac{\alpha \left(\gamma -1\right)}{V_{c}}\int p_{j}^{*}IdV}{1-\frac{\omega^{2}}{\omega_{J}^{2}}-i\frac{\omega }{\omega_{j}Q_{j}}}$$

In Formula (17), *ω_j_* represents the resonant angular frequency under normal mode *j*; *V*_C_ represents the cavity volume; *Q_j_* represents the quality factor under normal mode *j*; $p_{j}^{*}$ represents the conjugate complex number of sound pressure in simple positive mode *j*. In actual use, *ω* = *ω_j_* is usually used to ensure that the photoacoustic cell works in a simple positive mode, in which case, the sound pressure of the photoacoustic cell can be expressed as follows:(18)$$p\left(r_{M},\omega_{j}\right)=-(\gamma -1)\frac{Q_{j}}{\omega_{j}}\frac{L_{C}}{V_{C}}I_{j}p_{j}(r_{M})\alpha P_{0}$$

In Formula (18), *P_0_* represents the power of the light source; $I_{j}=\frac{1}{P_{0}C}\int p_{j}^{*}IdV$, and *C* represents the concentration of the gas; $-(\gamma -1)\frac{Q_{j}}{\omega_{j}}\frac{L_{C}}{V_{C}}I_{j}p_{j}(r_{M})$ is called the photoacoustic cell constant, expressed as *C*_cell_, which is a parameter related only to the structure, material, and size of the photoacoustic cell.

The sensitivity of the microphone is denoted as *M_s_*, and then, the photoacoustic signal *S_pas_* can be denoted as follows:(19)$$S_{pas}=M_{s}\cdot C_{cell}\cdot C\cdot \alpha \cdot P_{0}$$

According to Formula (19), when *M_s_*, *C*_cell_, *α*, and *P*_0_ are constant, the photoacoustic signal is proportional to the concentration of the gas to be measured, which is the theoretical basis of photoacoustic detection.

## 3. Construction of Photoacoustic Spectroscopy Experiment System

Figure 2 depicts the schematic structure of the photoacoustic HF gas-sensing system. A tunable distributed feedback (DFB) laser was used as an excitation light source. The DFB laser was driven by the current of modulating sinusoidal and scanning sawtooth waves. The light emitted by the DFB laser passed through the fiber collimator into the resonant photoacoustic cell. The HF gas in the photoacoustic cell absorbed modulated light to produce photoacoustic signals. The second-harmonic signal of HF gas absorption was obtained by using the phase-locked amplifier. An oscilloscope was used to collect the second-harmonic signal detected by the phase-locked amplifier.

A discharge chamber was designed. The actual corona discharge was simulated by the needle–plate electrode. Figure 3 shows the gas path connection and the electrodes in the schematic structure of the test system. To reduce the loss of HF in the gas cycle, each part of the structure was connected by pipes made of polytetrafluoroethylene. To reduce the gas flow noise in the experiment, a circulating pump (KVP04-1.1-24, Kamoer, Shanghai, China) and flowmeter (D07-7B, BEIJING SEVENSTAR FLOW CO., Ltd., Beijing, China) were used to maintain the gas flow rate at 50 mL/min. The pressure in the gas path was 0.3 MPa. The discharge chamber was made of quartz glass and stainless steel. The needle electrode was a high-voltage electrode. The volume of the discharge chamber was 2 L. The maximum discharge voltage was 60 kV. The vacuum pump (RV12, Edwards, Burgess Hill, UK) can reduce the air pressure of the gas path to 2 × 10^−3^ mbar.

### Absorption Lines of HF

The photoacoustic signal depended on the absorption intensity of the HF gas to the incident light. The absorption intensity of HF gas was not only related to the concentration of the gas and the length of the photoacoustic cell, but also to the intensity of the absorption spectral line of the gas. The absorption spectral line of HF was selected by querying the High-Resolution Transmission Molecular Absorption Database (HITRAN) [21].

Figure 4 shows the absorption lines of HF in the near-infrared region. It can be seen from Figure 4a that the absorption lines of HF in the near-infrared region were mainly concentrated in the bands of 1250–1350 nm and 2300–2800 nm. The absorption intensity of HF was greater in the mid-infrared region. Quantum cascade lasers can be used in the mid-infrared region, but they are too expensive. In addition, other gases also have strong absorption in the mid-infrared band, which will lead to interference. Therefore, the absorption line of 1312.59 nm was selected.

According to the HITRAN database, gas absorption lines of HF, H_2_O, CO, CO_2_, and H_2_S were obtained. Figure 5 is a comparison of the absorption intensity of HF and its main interference gases after amplification. As can be seen from Figure 5, HF had a strong absorption coefficient at 1312.59 nm, and there was no obvious interference from other gases. A DFB laser with a central wavelength of 1312.6 nm was selected as the excitation light source. According to the wavelength of the DFB laser, the collimator (50-1310A-APC, Thorlabs, Newton, NJ, USA) with the central wavelength of 1310 nm was selected.

There are two types of photoacoustic cells: non-resonant photoacoustic cells and resonant photoacoustic cells. Compared with non-resonant photoacoustic cells, resonant cells can detect flowing gas [22]. In this research, it was necessary to detect flowing gas, so resonant photoacoustic cells were selected. The choice of photoacoustic cell material is very important. The material is related to the damping, viscosity, and heat loss of the gas, which greatly affects the detection sensitivity of the photoacoustic system. Materials with high heat conductivity and a large Poisson ratio are generally selected for photoacoustic cells [23]. At the same time, the processing difficulty should be considered. Commonly used materials for photoacoustic cell are brass, copper, aluminum alloy, stainless steel, and so on. The characteristic parameters of these materials are shown in Table 1. Red copper has the highest thermal conductivity. However, its characteristic is soft, and it is difficult to ensure the smooth surface of the photoacoustic cell during processing. Among other materials, aluminum alloy has the highest thermal conductivity and Poisson ratio. At the same time, aluminum alloy is easy to process. Therefore, aluminum alloy was selected as the photoacoustic cell material.

The photoacoustic cells designed in this research adopted the first-order longitudinal resonance mode. Increasing the photoacoustic cell constant can improve the intensity of the acoustic signal. The photoacoustic cell constant can be increased by increasing the cavity length or decreasing the cavity radius. However, if the cavity radius is too small or the length is too large, it is easy to make light shine on the cell wall. Theoretical research shows that in order to achieve a better de-noising effect, the length of the buffer cavity should be 1/2 of the length of the resonant cavity, and the diameter of the buffer cavity should be more than 3 times the diameter of the resonant cavity [24]. In this study, the inlet and outlet were placed in the buffer chamber, which could effectively reduce the gas flow noise. The resonant chamber used in the test had a radius of 5 mm and a length of 100 mm. The buffer chamber was 50 mm long and 35 mm in diameter.

A photoacoustic cell window should be made of a material that does not react with HF, and the incident light transmission should be as large as possible. A WG51050 window was selected for use in this test. The material of WG51050 is CaF_2_, which does not react with HF. Its transmissivity in the near-infrared region is above 92%. According to the data provided by the manufacturer of THORLABS, the light transmission characteristic of WG51050 is shown in Figure 6.

A capacitive microphone has the advantages of high sensitivity, good transient characteristics, wide frequency response, and low cost. An electret microphone is a kind of capacitive microphone. An electret microphone provides polarization voltage by itself, which makes the internal circuit relatively simple, low cost, and small in size. The photoacoustic signal is sensitive to the shape of the photoacoustic cavity, so it is more appropriate to choose a small electret microphone. In this research, the MPA416 microphone made by Beijing Prestige Co., Ltd. (Beijing, China) was selected. The diameter of the microphone was 1/4 inch, the sensitivity was 50 mV/pa, and the frequency response range was 20 Hz~20 kHz. To determine the installation position of the microphone, COMSOL-Multiphysics 5.4 was used to simulate the frequency domain of the photoacoustic cell. The simulation results are shown in Figure 7. According to COMSOL calculation results, the first-order longitudinal resonance frequency of the photoacoustic cell was 650.4 Hz. As can be seen from Figure 7, when the resonant photoacoustic cell worked in the first-order resonance mode, the sound pressure at the middle position of the resonator was the largest. Therefore, the microphone needed to be installed in the middle of the resonator.

Based on the previous analysis, the main sizes of the designed photoacoustic cell are shown in Figure 8.

## 4. Experiment and Analysis

### 4.1. The Modulating Sinusoidal Wave Frequency of the DFB Laser

When the frequency of a modulating sinusoidal wave is half of the photoacoustic cell resonance frequency, the photoacoustic second-harmonic signal will be the strongest. Therefore, before the formal experiment, it was necessary to determine the resonance frequency of the photoacoustic cell.

To find the resonance frequency of the photoacoustic cell, the resonant frequency of the photoacoustic cell was tested using the system shown in Figure 2. In the test, H_2_S/SF_6_ mixed gas was used as the sample gas. The DFB laser (PL-DFB-1578-AA81-SA, LD-PD INC) had a central wavelength of 1578.12 nm. In order to enhance the photoacoustic signal, the output optical power of the DFB laser was amplified by using an Erbium-Doped Fiber Amplifier (EDFA). The output optical power of the EDFA was 1W. The peak-to-peak value of the modulating sinusoidal wave was 63.4 mV. In order to eliminate the influence of temperature change on the photoacoustic signal, the temperature control system was used to control the temperature of the photoacoustic cell at 30 °C.

The photoacoustic cell was filled with 100 μL/L H_2_S/SF_6_ standard gas. The gas pressure in the photoacoustic cell was 1 atmosphere. The amplitude of the modulating sinusoidal wave was 31.7 mV. The output power of the EDFA was set to 1 W. The frequency of the modulating sinusoidal wave was adjusted from 300 Hz to 345 Hz, and the value of the photoacoustic second-harmonic signal was recorded. The photoacoustic second-harmonic signal was fitted using a Lorentz curve. The fitting result is shown in Figure 9. As can be seen from Figure 9, when the frequency of the modulating sinusoidal wave was 321.8 Hz, the signal was the strongest. Therefore, the first-order longitudinal resonance frequency of the photoacoustic cell was 643.6 Hz.

### 4.2. Calibration of HF Concentration

To calibrate the detection sensitivity of the system, 525 μL/L HF/SF_6_ standard gas was used. Before the test, the photoacoustic cell was purged 5 times with the standard gas to passivate the aluminum alloy. The standard gas was quickly tested after intake, and the test time was 40 s. The test was repeated 12 times. The test results are shown in Figure 10. Figure 10 shows the mean and standard deviation of each test.

As can be seen from Figure 10, the photoacoustic signal gradually increased and became stable with the increase in the number of test times. This was due to the passivation of aluminum alloy to reduce the loss of HF. After 12 tests, the photoacoustic signal no longer increased. The result of the 12th test was used to calculate the detection sensitivity of the system to HF gas. Figure 11 shows the photoacoustic second-harmonic wave of the 12th test. The mean value of the photoacoustic signal in the 12th test was 233.750 μV. By calculation, the detection sensitivity of the test system was 0.445 μV/(μL/L).

The *LOD* is a very important parameter used to measure the ability of detection systems. It represents the lowest concentration of gas that can be detected by the system. The *LOD* can be expressed by Formula (20).
(20)$$LOD=\frac{K\sigma_{s}}{S}$$

In Formula (20), *K* represents the confidence coefficient, *σ_s_* represents the standard deviation of the signal, and *S* represents the sensitivity of the detection system. According to the detection result, the standard deviation of the signal was 0.272 μV. When *K* is 1, *LOD* is 0.611 μL/L according to Formula (20).

### 4.3. Online Detection of HF under Simulated GIS Corona Discharge Fault

The gas tightness of the detection system was tested. Vacuum and 0.1 atm (gauge pressure) positive pressure gas tightness tests were carried out on the gas path system. Under both test conditions, there was no change in gas pressure after 12 h. This showed that the gas system was well sealed. Before each experiment, the discharge room and electrodes were scrubbed with anhydrous ethanol. Then, the gas system was connected and the gas system was cleaned five times using the vacuum pump and SF_6_ standard gas. Finally, the SF_6_ standard gas was inflated into the discharge electrical room to make the gauge pressure reach 0 atm.

Under the action of the circulating pump, the gas in the discharge room circulated in the gas path. The photoacoustic second-harmonic detection technology and wavelength modulation technology were used to test the HF concentration in the gas path. The peak-to-peak value of the scanning sawtooth wave of the DFB laser was 50 mV, and the frequency was 0.1 Hz. The peak-to-peak value of the modulating sinusoidal wave of the DFB laser was 50 mV and the frequency was 321.8 Hz.

The actual GIS corona discharge fault was simulated by the needle–plate electrode discharge. The photoacoustic spectroscopy test system shown in Figure 2 was used for the online detection of HF generated during discharge. The plate electrode was copper, and the needle electrode material was aluminum alloy. The discharge voltage was 20 kV. The discharge lasted for 372 min. During the discharge, HF was also reduced due to chemical reactions with materials in the gas path. The concentration of HF in the gas path over time during the discharge is shown in Figure 12. As can be seen from Figure 12, from 0 to 150 min, the concentration of HF rose faster. This is because the HF produced by the discharge was much more than that consumed. From 150 min to 264 min, the concentration of HF was saturated. This is because the HF produced by the discharge was approximately equal to that consumed. From 264 min to 372 min, the concentration of HF showed a downward trend. This may have been due to the weakening of the discharge energy caused by the ablation of the aluminum alloy needle electrode in the discharge, resulting in less HF being produced by the discharge than consumed.

When the voltage was reduced to 0, the discharge stopped. The concentration of HF over time is shown in Figure 13. As can be seen from Figure 13, the concentration of HF decreased approximately linearly. This was due to the reaction of HF with materials in the gas path, resulting in a reduction in its concentration. Therefore, offline detection cannot reflect the HF concentration when a GIS fault occurs. In order to accurately measure HF concentrations in GIS, online monitoring technology is necessary.

## 5. Conclusions

In this research, the concentration of trace HF gas produced by GIS corona discharge was detected. An online detection platform was built, which could realize the real-time online detection of HF gas concentrations. Compared with previous studies, the platform was more consistent with real corona discharge characteristic gas-detection scenarios, which provided an important basis for GIS condition assessment. A detection system for HF gas online monitoring was built. A gas circuit system with good sealing was designed. A corona discharge fault in GIS was simulated by using needle–plate electrode discharge. Gas photoacoustic spectroscopy was used to detect HF generated in discharge in real time. In order to detect flowing gas, a resonant photoacoustic cell was designed. According to the mechanism of photoacoustic signal generation, the material of the photoacoustic cell and the size of the photoacoustic cavity were reasonably designed. By querying the HITRAN database, the absorption intensities of HF and interference gases were compared, and the absorption spectral line of HF in the near-infrared region was determined. Wavelength modulation spectroscopy and second-harmonic detection techniques were used in the detection. The online monitoring of HF generated in a simulated GIS corona discharge fault was successfully realized by using the test system. The test system was calibrated using 525 μL/L standard HF/SF_6_ mixed gas. The detection sensitivity of the test system for HF gas was 0.445 μV/(μL/L), and the *LOD* was 0.611 μL/L.

HF was found to be reduced by reacting with materials in the gas path. In the early stage of a discharge fault, the HF concentration increases rapidly. With the ablation of the discharge point, the discharge energy may weaken, resulting in a decrease in the concentration of HF. The research results of this paper have important significance in GIS online monitoring.

## Figures and Tables

**Figure 1 sensors-24-02806-f001:**
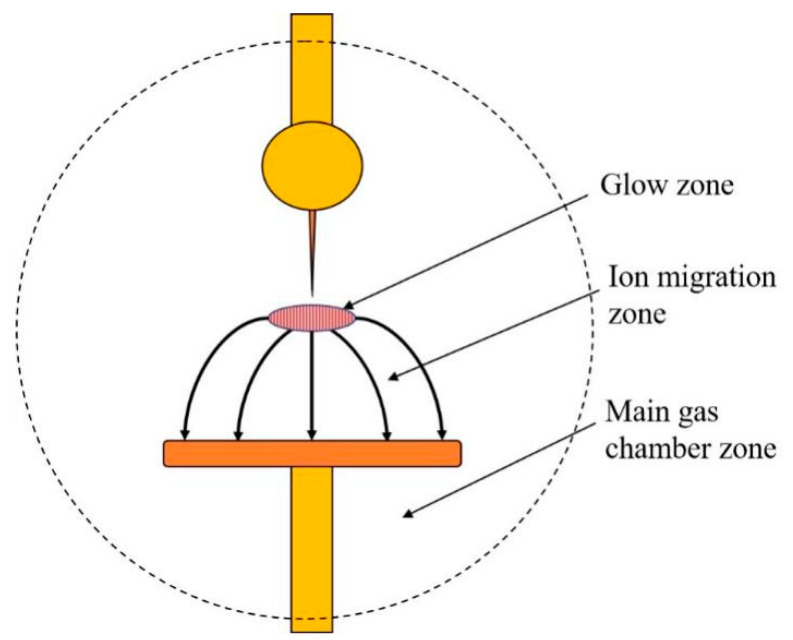
Three-zone model of gas discharge.

**Figure 2 sensors-24-02806-f002:**
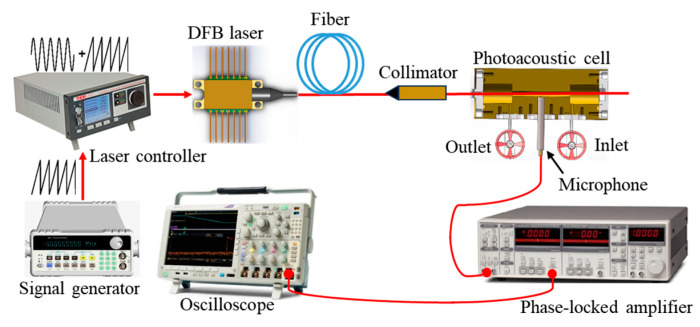
Schematic structure of the test system for the photoacoustic trace HF sensor.

**Figure 3 sensors-24-02806-f003:**
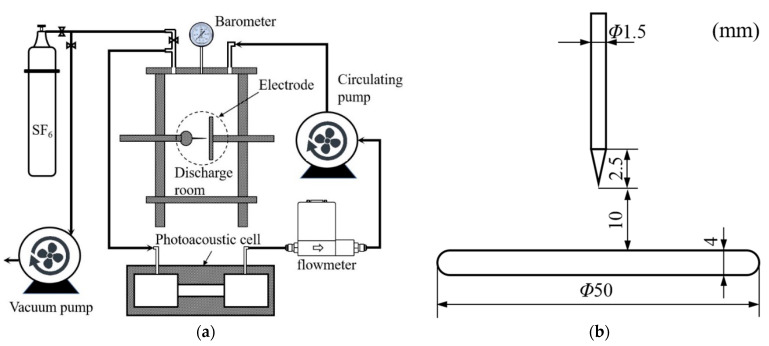
Schematic structure of the gas path structure of the test platform and electrode. (**a**) Gas path connection, (**b**) Electrode.

**Figure 4 sensors-24-02806-f004:**
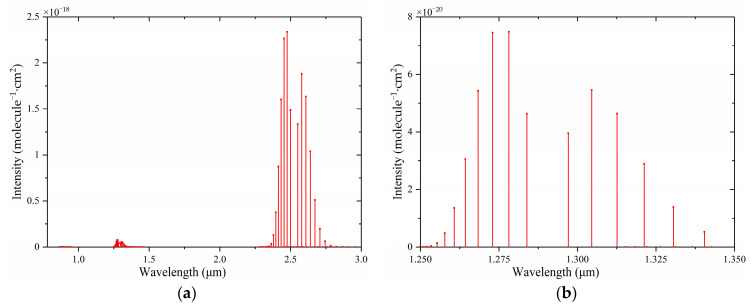
Absorption lines of HF in the near-infrared region: (**a**) 0.78~3 μm, (**b**) 1.25~1.35 μm.

**Figure 5 sensors-24-02806-f005:**
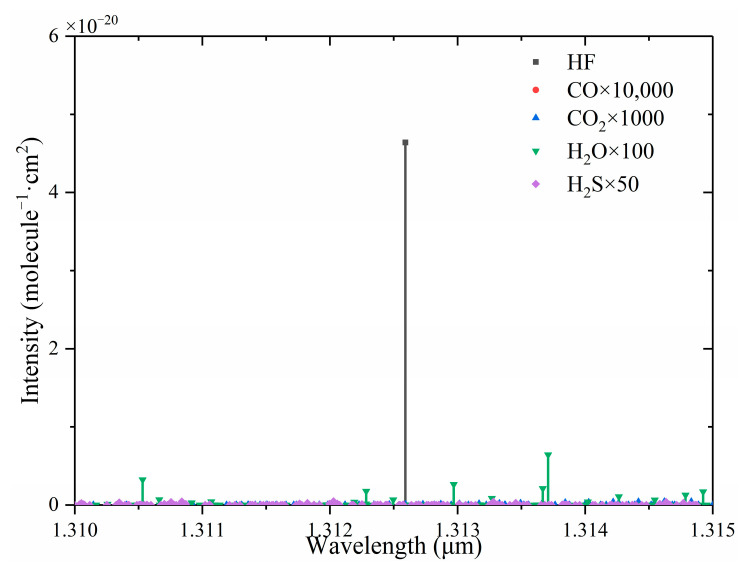
Absorption lines of HF and its main interfering gases near 1310 nm.

**Figure 6 sensors-24-02806-f006:**
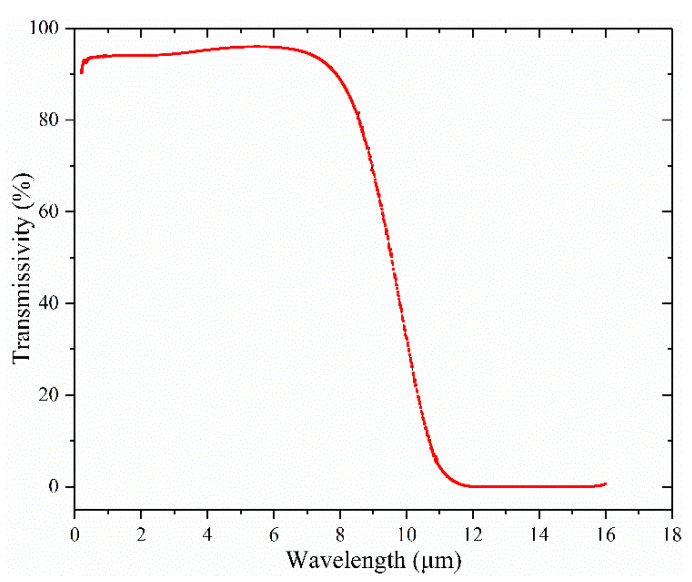
Transmissivity characteristic curve of CaF_2_ window.

**Figure 7 sensors-24-02806-f007:**
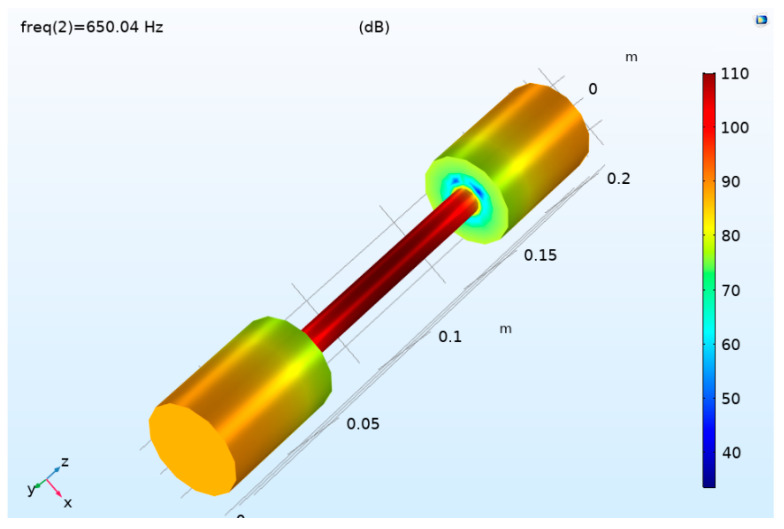
Simulation results of sound field distribution in photoacoustic cell.

**Figure 8 sensors-24-02806-f008:**
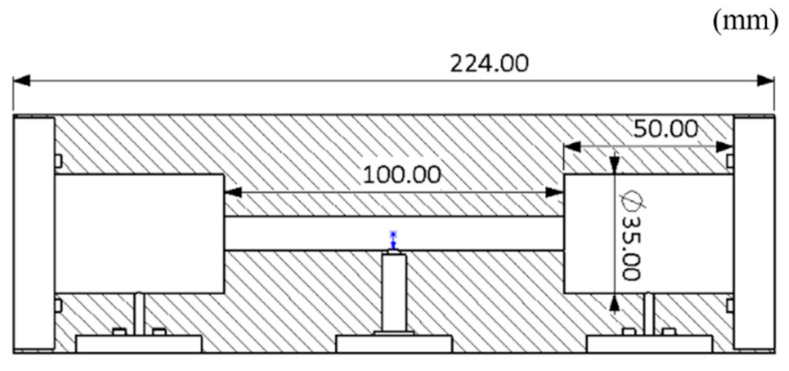
Sizes of the photoacoustic cell.

**Figure 9 sensors-24-02806-f009:**
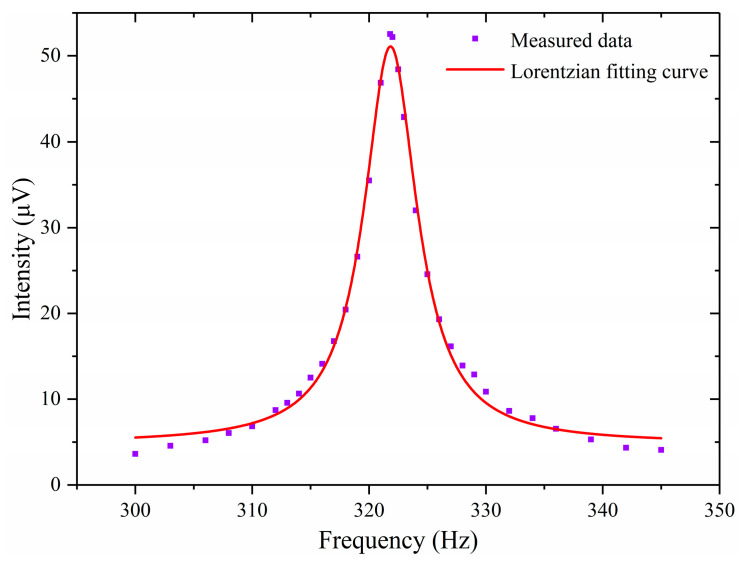
Frequency response curve of photoacoustic cell.

**Figure 10 sensors-24-02806-f010:**
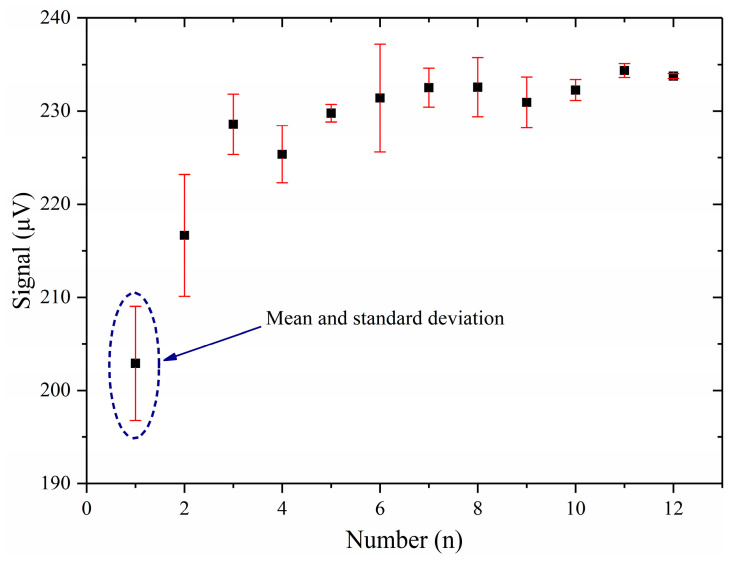
The test result of the system for the standard gas.

**Figure 11 sensors-24-02806-f011:**
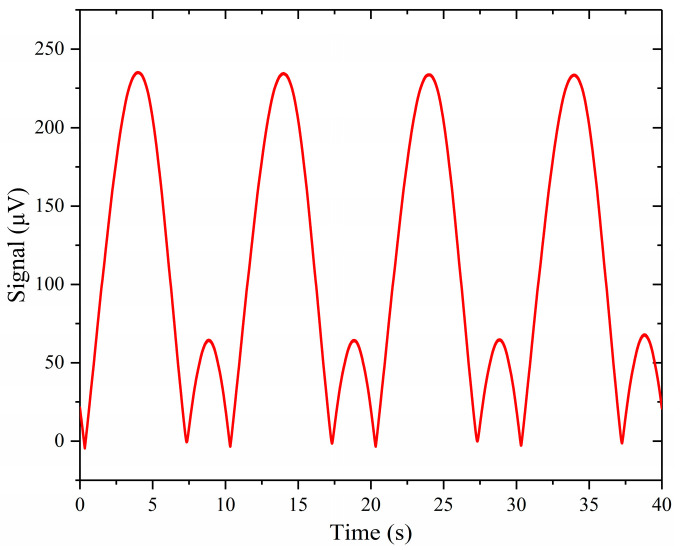
The photoacoustic second-harmonic wave of the 12th test of the system for the standard gas.

**Figure 12 sensors-24-02806-f012:**
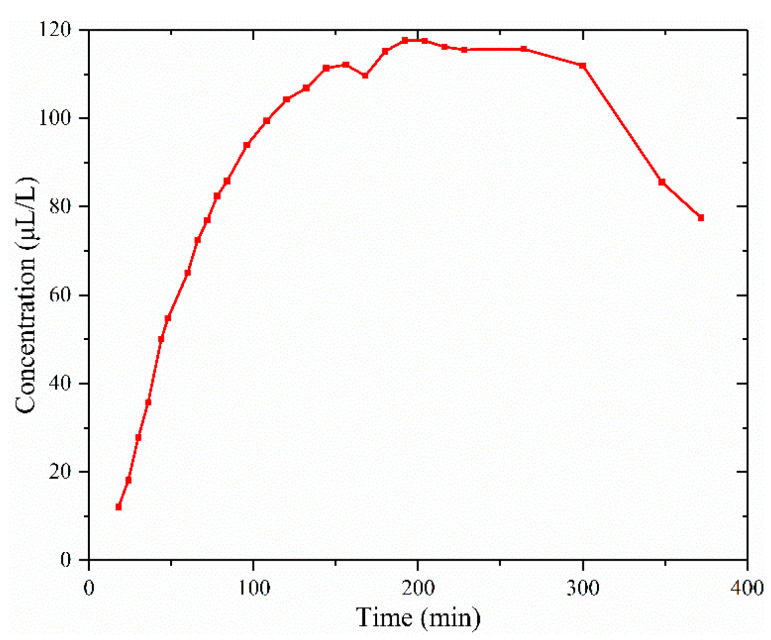
HF concentration curve with discharge voltage of 20 kV.

**Figure 13 sensors-24-02806-f013:**
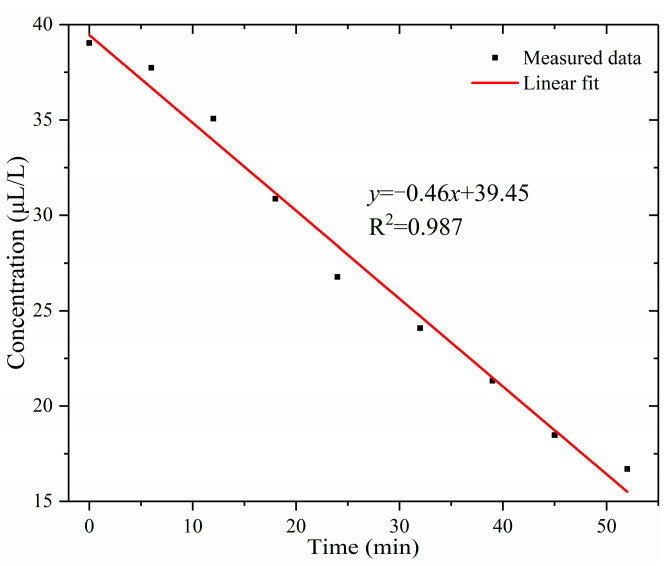
Variation in HF concentration with time after the discharge stopped.

**Table 1 sensors-24-02806-t001:** Characteristic parameters of materials commonly used in photoacoustic cells.

Material	Density (g·cm^−3^)	Thermal Conductivity (W·m^−1^·K^−1^)	Poisson Ratio (μ)
Copper	8.43	108.9	0.32
Red copper	8.92	386.4	0.35
Aluminum alloy	2.90	167	0.34
Stainless steel	7.93	16.2	0.30

## Data Availability

All data underlying this article are available in the article.

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
