# Peer review of "Online Detection of Hydrogen Fluoride under Corona Discharge in Gas-Insulated Switchgear Based on Photoacoustic Spectroscopy"

_sensors, 2024, doi:10.3390/s24092806_

Round 1
Reviewer 1 Report
Comments and Suggestions for Authors
In this manuscript, wavelength modulation and second harmonic detection technology were adopted to online detect HF. In general, the quality of the work is average. Some of the most important contents are required to elaborate the working principle and evaluate the real performance of the experiment. It is also highly recommended that the manuscript should be reviewed by English correction services. In my opinion, the paper should not be accepted until the following concerns have been thoroughly resolved.
1. The methods or technologies mentioned in this paper, such as second harmonic, wavelength modulation, etc., are mature and widely used technologies, so what are the innovation points in this paper.
2. Many symbol errors in the manuscript, please check carefully. Lines 121, lines 129, lines 131, lines 132, lines 166, lines 181, lines 221, 222…….
3. Some necessary references are not cited. Lines 167…
4. In lines 195, lines 206-209, lines 209-213, lines 226-227, the authors make strong affirmations without proper reference.
5. Figure 7 shows the acoustic pressure distribution of the photoacoustic cell simulated by COMSOL, why not add the frequency response characteristic curve to the manuscript.
6. Lorentz fitting should be adopted and calculate the relevant parameters in figure 10 and calculate the relevant parameters.
7. In section 4.2, the authors only measured a set of concentration gases to calibrate the linearity of the system, and the amount of data was insufficient to support the conclusion. At the same time, the system performance is not analyzed concerning the noise level.
8. To promote the quality and format of the images, the authors are advised to review your reference 19, carefully.
9. Please check figure 2 carefully. The collimator of this kind of system structure is generally located on one side of the buffer cavity of the photoacoustic cell, and the position marked in the figure is the optical fiber. And some words are occluded.
10. Units need to be preceded by spaces, please refer to the template of the journal.
11. Some mathematical symbols are not formatted correctly, please check carefully.
12. The resolution of the picture in this article is insufficient, please increase the resolution to at least 300 dpi.
13. Figure 2 and Figure 9 are the same, only EDFA difference, please consider deleting picture 9.
14. The description of the instrument or equipment is incorrect, please refer to the template of the journal for thorough revision.
15. Formulas need to be written in a consistent format.
16. Please check the writing tenses carefully.
Comments on the Quality of English LanguageIt is also highly recommended that the manuscript should be reviewed by native English speakers or by English correction services.
Reviewer 2 Report
Comments and Suggestions for Authors
Overall the paper is well written and illustrated for this novel design of online HF detection.
There are several sections still needs to be better discussed or explained. One of the major issue is that the references are not well organized and more references may need to support the author’s statements. Another main drawback is the results are not discussed adequately, especially not correlated to the results of the current established methods, which will weaken the novelty and convincement of the conclusions by the authors. The detailed question/suggestion list is here:
1. Page 1, line 16, what’s the GIS stand for? Should not use abbreviation at the first time.
2. Page 3, line 72, reference for the standard IEC60480-2020.
3. Page 3, line 82, it would be better to put a reference, i.e. a review, for the statement here.
4. Page 5, line 144, figure 3b, what’s the unit of the diagram?
5. Page 6, section 3.1, please make it clear that the chosen absorption lines are selected according to the HITRAN database or based on your experimental results (especially the origin of figure 4 and 5). If they are chosen from database, please add the reference accordingly, and if they are from your own experiments, please specific experimental conditions and a discussion of the optimization of this condition.
6. Page 7, figure 7, the cutoff wavelength information needs a reference or experimental conditions.
7. Page 10, section 4.2, one suggestion is to make a calibration curve with a series of concentration ratio of HF/SF6 standards, in this way the calculated limit of detection would be more accurate. You can also discuss the difference between the calibration curve and the 525uL/L single point.
8. Page 12, before the conclusions section, please compare the current results in this paper with the previously report data and discuss the reason, impact and novelty of the difference.
9. Some minor text issues like CaF2, H2O… The numbers need to be subscripted.
Please review the comments and address the response accordingly, thanks.
Round 2
Reviewer 1 Report
Comments and Suggestions for Authors
There are no comments or suggestions for authors.